# Risk perception, community myth, and practices towards COVID-19 pandemic in Southeast Ethiopia: Community based crossectional study

**Ahmednur Adem Aliyi** [1]*, **Musa Kumbi Ketaro**[1], **Zinash Teferu Engida**[1], **Ayele Mamo Argaw** [2], **Abduljewad Hussen Muhammed**[1], **Mesud Mohammed Hassen**[2], **Abdushekur Mohammed Abduletif**[3], **Damtow Solomon Shiferaw**[4], **Abate Lette Wodera**[1], **Sintayehu Hailu Ayene**[1], **Jeylan Kassim Esmael**[1], **Edao Sinba Etu**[1]

1 Department of Public Health, School of Health Science, Goba Referral Hospital, Madda Walabu University, Goba, Ethiopia, 2 Department of Pharmacy, School of Health Science, Goba Referral Hospital, Madda Walabu University, Goba, Ethiopia, 3 Department of Nursing, School of Health Science, Goba Referral Hospital, Madda Walabu University, Goba, Ethiopia, 4 Department of Anatomy, School of Health Science, Goba Referral Hospital, Madda Walabu University, Goba, Ethiopia

* ahmedhariro@gmail.com

**Data Availability Statement:** All relevant data are within the paper and its Supporting information files.

## Abstract

### Objective

The objective of this study was to assess risk perception, community myths, and preventive practice towards COVID-19 among community in Southeast Ethiopia, 2020.

### Methods

Community-based cross-sectional study was conducted among 854 participants selected using a multistage sampling technique. Data were collected using a structured questionnaire adapted from previous literature. Descriptive statistics were done to summarize the variables. A generalized linear model with binary logistic specification was used to identify factors associated with risk perception and practice. Accordingly adjusted odds ratios with 95% confidence intervals were calculated and those with p-value < 0.05 were considered as significant factors associated with risk perception and practice. Cluster analysis using a linear mixed model was performed to identify factors associated with community myth and those with p-value <0.05 were reported as significant factors associated with community myth.

### Results

All 854 respondents gave their answer yielding 100% response rate. Of these 547 (64.1%) were male, 611 (71.5%) were rural residents, 534 (62.5%) got information about COVID-19 from TV/radio, 591 (69.2%) of them live near health facility, 265 (30.8%) have a history of substance use and 100 (11.7%) have a history of chronic illness, and 415 (48.6%) of them have a high-risk perception, 428 (50.1%) have a wrong myth about COVID-19 and 366

**Funding:** The authors received no specific funding for this work.

**Competing interests:** The authors have declared that no competing interests exist.

(42.9%) have poor practice respectively. Residence, distances from health facility and myths were significantly associated with risk perception. Occupation, knowledge, and practice were significantly associated with community myths. Also level of education, living near health facilities, having good knowledge and wrong myth were significantly associated with the practice of utilizing COVID-19 preventive respectively.

## Conclusion

The study found high-risk perception, high wrong community myth, and relatively low utilization of available practices towards COVID-19 and factors associated with them.

## Introduction

COVID-19 was initially started in December 2019, and later it was stated as pandemic and has been declared as a Public Health Emergency of International Concern by the WHO [1]. Approximately, 20.0% of COVID-19 patients developed severe symptoms, which included respiratory and bleeding disorders [2]. The virus can also be transmitted through the respiratory tract of patients with signs and symptoms but can also transmit from asymptomatic individuals before the onset of clinical features [3]. Susceptibility to COVID-19 looks to be associated with sociodemographic characteristics such as low education, age, and low access to information as well as with underlining co-morbidities like diabetic mellitus, cancer, and chronic respiratory illnesses [4].

The contagious COVID-19 virus outbreaks needs a urgent response from all stakeholders and communities [5]. Increasing public awareness and working in collaboration with the communities has endless benefits in curbing this pandemic [6]. People's risk perception of the pandemic affects the utilization of available preventive measures [7]. The true risk from the COVID-19 virus might be low, but it gets media attention and become the candidate of social media discussion, which might have effects on risk perception, which in turn may determine communities' behavior in adopting and using these pandemic preventive measures [8]. Understanding the risk perception among people was crucial to understand ways of delivering information for communities using correct information channeling [5, 9]. Few studies conducted previously reveal conflicting findings on the level of perceived risk towards Novel Coronavirus. One survey conducted in Italy identified the effect of age on risk perception and recommend importance of delivering correct information about the disease and its prevention mechanism [10].

The others studies conducted in Iran found moderate risk perception in the community [11, 12]. Opposing to the above two, the study conducted in United States reveals a low level of risk perception [13]. Perceived risk differs across different sociodemographic characteristics including age, educational level, residence, and access to information [9, 11, 14]. The study conducted on COVID-19 risk perception in Vietnam also identified the effect of using social media on risk perception towards COVID-19 [5].

Misinformations that transmit from different media also start helping the effect of this pandemic in affecting people's behavior towards it. This leads to the development of some popular myths like "COVID-19 doesn't exist at all", "it can't affect people in the hot or cold environment" and "COVID-19 was deliberately created by the people". These myths have good and bad consequences on health [15]. These bulk of the information which is circulating through multiple channels influences how people think about the disease and their readiness to stick to

available preventive methods [5]. This mandate as channeling of basic information should be from trusted sources. Various studies identified the level of utilizing available COVID-19 pandemic preventive techniques and variability of using these methods across different sociodemographic and socioeconomic characteristics [11, 16, 17]. Hence understanding level of risk perception towards the COVID-19 pandemic, identifying myth developed in the communities following this pandemic, and their utilization of available preventive measures has crucial importance in reducing COVID- 19 transmission. Therefore this study was conducted to identify risk perception, community myth, and practice towards COVID-19 pandemic in Southeast Ethiopia.

## Methods

### Study setting, design, and period

A community-based crossectional study was conducted from March to June 2020 among 854 adult populations who were permanent residents of 22 Kebeles in two Zones of the Oromia region, Southeast Ethiopia.

### Inclusion and exclusion criteria

Respondents with an age greater than 18 years were included. Any individual who was not a permanent resident of the study area, critically ill, with hearing impairment, and has changes in consciousness level or cognitive disorders were excluded from the interview.

### Sample size determination

Single population proportion formula for sample size determination was used to calculate the required sample size by taking the proportion of participants with high-risk perception against COVID-19 50%; at 95% confidence level, 5% margin of error, 10% non-respondents, and design effect of 2. This gave the final sample size of 854 individuals.

$$n = \frac{(z \propto /2)2\, p(1-p)}{d2}$$

After multiplying by design effects = 768. Then 10% of non -respondent rate was added and the overall sample size became 854.

### Sampling procedure and data collection

A multistage sampling technique was used in which woredas and administrative towns were selected after grouping. From selected woredas and administrative towns 22 kebeles (villages) were selected by using lottery methods. Then 854 participants were randomly included from systematically selected households. Data regarding risk perception, community myth, and preventive practice against COVID-19 were collected using tools adapted from previous studies and WHO recommendations [2, 5, 13, 18–24]. Data regarding sociodemographic characteristics and source of information about COVID-19 were collected by using tools adapted from EDHS and previous articles [9, 16, 19, 25]. The questionnaire was initially prepared in English and then translated into Afan Oromo. Translation back to English was done to check for consistency by languages experts. A questionnaire pretest was done before actual data collection on 5% of the sample size. The questionnaire was modified based on pre-test results.

Data were collected by trained data collectors. Two days training was given to the data collector on objectives, relevance of the study, confidentiality, respondent right, informed consent, and on actual data collection procedures. Ethical clearance from Madda Walabu

University Research and publication and letter of permission from selected woredas and administrative towns were obtained. After a brief description of the objectives of the study to every study participant oral consent was obtained. Then questionnaires were administered face to face by the data collectors. This is the appropriate approach for people with no formal education. During data collection data collectors gives clarification to the questions misunderstood by respondents. Consistency and completeness of data were checked by investigators every day. After data collection, filled questionnaires were kept carefully.

## Variables measurements

Risk perceptions of respondents were assessed by asking six questions adapted from previously conducted studies [11, 26, 27], Total risk perception score was computed by adding individual responses of these six questions. Then median score was used to categorize the level of risk perception. Respondents those score less than the median were categorized as having low-risk perception, and those scores equal to or above the median were categorized as having high-risk perception regarding the COVID-19 pandemic.

Myths about COVID-19 pandemic were also measured by asking six questions adapted from previous studies [21, 22, 28]. The total myth score was calculated by adding responses to these six questions. The median score was used to label individuals as with wrong myth and not with wrong myth. Accordingly, those scores less than the median were categorized as having no wrong myth, and those scores equal to or greater than the median were categorized as having wrong myth.

Regarding utilization of preventive practice towards COVID-19; respondents were asked twelve questions adapted from World Health Organization advice to the public and previous studies [29–32]. The total practice scores was computed by adding responses to these questions. And median score was used to categorize practice of participants. Accordingly practice of respondents regarding utilization of COVID-19 pandemic preventive measures was categorized as having poor practice and good practice based on their median score computed from these twelve asked questions. Those with a score below the median were categorized as having poor practice and those with a score greater than or equal with median were categorized as having good practice. Refer to appendix one for the questionnaire. The data about sociodemographic variables, access to health care, and source of information were collected by adapting tools from EDHS 2016 and previous studies after some modification.

## Data processing and analysis

Data were checked for completeness and entered to Epidata version 3.1 and were exported to SPSS version 25 for analysis. Data cleaning was done using frequency distribution and descriptive statistics. The scores for risk perception, community myth and practice in utilization of COVID-19 were computed from their respective individual questions responses. Sociodemographic characteristics, access to health care and source of information were summarized using frequency distribution. Average values of all outcomes were calculated and reported. The scores of risk perception, community myth, and preventive practice were compared across different sociodemographic characteristics of respondents. A generalized linear model was used to examine factors associated with risk perception and practice regarding the utilization of available COVID-19 preventive measures. Adjusted odds ratio with a 95% confidence interval was computed and those with a p-value less than 0.05 were reported as significant factors associated with risk perception and practice. Cluster analysis by using a linear mixed model was performed to identify factors associated with community myth. Variables with a p-value

less than 0.05 in the linear mixed model were reported as significant factors associated with community myth.

# Results

## Sociodemographic characteristics of respondents

As shown in "Table 1" below 854 respondents participated in this study yielding 100% response rate. Out of the total 854 respondents, 547 (64.1%) were male and 845 (98.9%) were Oromo ethnicity, 611 (71.5%) were rural residents. Regarding occupational status, 499 (58.4%) were farmers, the roles of 660 (77.3%) were father/mother and the majority of them attends primary level of education 335 (39.2%). Concerning their marital status, 645 (75.5%) were married earning the median monthly income of 1675.27 ETB. Around two-thirds of the respondents 534 (62.5%) got information about COVID-19 from TV/radio, 591 (69.2%) of them live near health facility and 265 (30.8%) have a history of substance use mostly khat 228 (26.7%) and 100 (11.7%) of them have a history of chronic illness. Finally, more than two-thirds of the total participants 604 (70.7%) live in their own house.

## Distribution of risk perception, community myth and preventive practices towards COVID-19 in communities, 2020 (n = 854)

Risk perception towards COVID-19 was computed from six questions. Its median score is 19 ranging from six to thirty. Those who have risk perception greater than the median score were classified as having high-risk perception. Accordingly around half of the study participants (415 (48.6%) have high-risk perceptions. Risk perception is the same across gender and residence but comparably higher among those who live near health facilities, non-governmental workers, and have a history of chronic illness.

Community myth was assessed by asking six questions. Based on this 428 (50.1%) of the study participants have the wrong myth about the COVID-19 pandemic. The median community myth was higher among females and urban residents. Concerning its distribution across occupation types, it was higher among NGO workers and the lowest among farmers. Also, the lower community myth was scored among less educated and those with a history of substance use. It was also relatively higher among those with a history of chronic illness.

The scores for practice towards utilization of COVID-19 pandemic was computed from 12 questions and the median score was used to categorize participants. Of all participants, 366 (42.9%) of them had low utilization of the stated preventive practice.

There is no gender difference in using practice to prevent COVID-19. Using these preventive practices was higher among urban residents and those near to health facilities but relatively lower among farmers, those without formal education, and substance users. See "Table 2" below.

## Factors associated with risk perception, community myth and practices towards COVID-19 in communities, 2020 (n = 854)

**1. Factors associated with risk perception towards COVID-19 pandemic.**   Seven variables were found to be eligible for multivariable generalized linear model analysis based the results from bivariate output. These are residence, level of education, distance from the health facility, history of chronic illness, knowledge about COVID-19, myth in the community, and status of utilizing COVID-19 preventive practice.

In the final multivariable generalized linear model three factors (variables) were found to be significantly associated with risk perception towards the COVID-19 pandemic. Accordingly

**Table 1. Sociodemographic characteristics of study participants.**

| Variables | Frequency (n) | Proportion (%) |
|---|---|---|
| **Gender** | | |
| Male | 547 | 64.1 |
| Female | 307 | 35.9 |
| **Age in years** | | |
| Mean | 34.12 | |
| Standard deviation | 13.29 | |
| Minimum | 16 | |
| Maximum | 90 | |
| **Ethnicity** | | |
| Oromo | 845 | 98.9 |
| Amhara | 4 | 0.5 |
| Tigre | 3 | 0.4 |
| Others | 2 | 0.2 |
| **Residence** | | |
| Rural | 611 | 71.5 |
| Urban | 243 | 28.5 |
| **Occupation** | | |
| Government workers | 113 | 13.2 |
| NGO employee | 7 | 0.8 |
| Private workers | 161 | 18.9 |
| Farmers | 499 | 58.4 |
| Others | 74 | 8.7 |
| **Role in family** | | |
| Father/Mother | 660 | 77.3 |
| Son | 143 | 16.7 |
| Daughter | 51 | 6 |
| **Level of education** | | |
| No formal education | 213 | 24.9 |
| Primary education (1–8) | 335 | 39.2 |
| Secondary education (9–12) | 178 | 20.8 |
| College and above (12+) | 128 | 15 |
| **Marital status** | | |
| Married | 645 | 75.5 |
| Single | 183 | 21.4 |
| Divorced | 24 | 2.8 |
| Others | 2 | 0.2 |
| **Average monthly income ETB** | | |
| Median | 1675.27 | |
| Minimum | 10 | |
| Maximum | 50000 | |
| **Source of information** | | |
| Religious leaders | 157 | 18.4 |
| TV/radio | 534 | 62.5 |
| Social media | 285 | 33.4 |
| Health workers | 55 | 6.4 |
| Gov't announcements | 74 | 8.7 |
| Another source (like telecom) | 1 | 0.1 |

(*Continued*)

**Table 1.** (Continued)

| Variables | Frequency (n) | Proportion (%) |
|---|---|---|
| **Distance from health facility** | | |
| Near health facility | 591 | 69.2 |
| Long-distance > 1 hour | 256 | 30 |
| Don't know | 7 | 0.8 |
| **Substance use** | | |
| No | 589 | 69 |
| Yes | 265 | 31 |
| **Types of substance used** | | |
| Khat | 228 | 26.7 |
| Cigarettes smoking | 33 | 3.9 |
| Alcohol drinking | 28 | 3.3 |
| Other substance | 1 | 0.1 |
| **History of chronic illness** | | |
| No | 754 | 88.3 |
| Yes | 100 | 11.7 |
| **House ownership** | | |
| No | 250 | 29.3 |
| Yes | 604 | 70.7 |

residence, distances from health facility and underlining myths were significantly associated with risk perception. Hence being a rural resident increase the likelihood of having high-risk perception by 2.4 (AOR 2.4 with 95%CI 1.67, 3.43) when compared to urban residents, living near health facility also increase the likelihood of having high-risk perception by 1.48 (AOR 1.48 with 95% CI 1.06, 2.06) when compared with those live far from health facility as well as having wrong myth also increase the likelihood of having risk perception by 1.39 (AOR 1.39 with 95% CI 1.2, 1.9) when compared to those without wrong myth. See "Table 3" below.

**2. Factors associated with community myth regarding COVID-19 pandemic.** Cluster analysis by using linear mixed model was used to identify factors associated with community myth. Accordingly seven variables were selected for final cluster analysis using a linear mixed model based on bivariate analysis results. These were gender, occupation, residence, distance from the health facility, knowledge regarding COVID-19, level of risk perception, and status of practice regarding utilization of COVID-19 preventive techniques.

In the final multivariable model three variables were found to be significantly associated with community myth. Accordingly being NGO employees, knowledge regarding COVID-19 and status of utilization of COVID-19 preventive techniques were significantly associated with community myth. Being an NGO employee positively related to community myth while poor knowledge regarding COVID-19 and poor utilization of available COVID-19 preventive techniques were negatively associated with the average score of community myth after controlling for the effects of other variables in the model. See "Table 4".

**3. Factors associated with practice towards utilization of COVID-19 preventive measures.** Generalized linear model was used to identify factors associated with practice towards utilization of COVID-19 preventive measures. Nine variables were found to be eligible for multivariable generalized linear model analysis by using results from the bivariate analysis.

These are gender, age, education, distance from the health facility, substance use, knowledge about COVID-19, underlining myth, the existing level of risk perception, and monthly

**Table 2. Distribution of risk perception, community myth, and preventive practices towards COVID-19.**

| Variables | Frequency(n) | Percent(%) |
|---|---|---|
| **Risk perception** | | |
| Minimum | 6 | |
| Maximum | 30 | |
| Median | 19 | |
| **Myth** | | |
| **Not all people develop a severe condition** | | |
| No | 511 | 59.8 |
| Yes | 343 | 40.2 |
| **Eating/contacting wild animals could cause COVID-19** | | |
| No | 368 | 43.1 |
| Yes | 486 | 59.6 |
| **Traditional medicine could prevent/cure COVID-19** | | |
| No | 682 | 79.9 |
| Yes | 171 | 20 |
| **COVID-19 was deliberately created by people** | | |
| No | 796 | 93.2 |
| Yes | 58 | 6.8 |
| **Living in a hot/cold environment could prevent COVID-19** | | |
| No | 763 | 89.3 |
| Yes | 91 | 10.7 |
| **COVID-19 could be transmitted by mosquitoes/housefly** | | |
| No | 631 | 73.9 |
| Yes | 223 | 26.1 |
| Total myth score | 1369 | |
| Minimum myth score | 0 | |
| Maximum myth score | 6 | |
| Median myth score | 2 | |
| **Practice** | | |
| **Going to crowded places** | | |
| No | 429 | 50.2 |
| Yes | 425 | 49.8 |
| **Stay at home** | | |
| No | 364 | 42.6 |
| Yes | 490 | 57.4 |
| **Reduce consuming outdoor foods** | | |
| No | 306 | 35.8 |
| Yes | 547 | 64.1 |
| **Avoid handshaking** | | |
| No | 208 | 24.4 |
| Yes | 646 | 75.6 |
| **Reduce public transportation** | | |
| No | 386 | 45.2 |
| Yes | 468 | 54.8 |
| **Frequently wash hands** | | |
| No | 164 | 19.2 |
| Yes | 690 | 80.8 |
| **Pay more attention to personal hygiene** | | |

(*Continued*)

**Table 2.** (Continued)

| Variables | | Frequency(n) | Percent(%) |
|---|---|---|---|
| | No | 176 | 20.6 |
| | Yes | 678 | 79.4 |
| **Use disinfectants** | | | |
| | No | 482 | 56.4 |
| | Yes | 372 | 43.6 |
| **Use face mask** | | | |
| | No | 148 | 17.3 |
| | Yes | 706 | 82.7 |
| **Maintain safe social distance** | | | |
| | No | 224 | 26.2 |
| | Yes | 630 | 73.8 |
| **Practicing respiratory hygiene** | | | |
| | No | 150 | 17.6 |
| | Yes | 704 | 82.4 |
| **Avoid touching eyes, nose and mouth** | | | |
| | No | 296 | 34.7 |
| | Yes | 558 | 65.3 |
| **Total (sum) of practice score** | | 6904 | |
| **Minimum practice score** | | 0 | |
| **Maximum practice score** | | 12 | |
| **Median practice score** | | 9 | |

income. After running a multivariable generalized linear model four variables were found as factors significantly associated with practice towards utilization of COVID-19 preventive measures. Accordingly having educational status of college and above increases the likelihood of practicing COVID-19 preventive techniques by 4.25 and 2.16 (AOR 4.25 with 95% CI 2.35, 7.69 and 2.16 with 95% CI 1.30, 3.63) when compared with those with no formal education and primary education respectively. Living near a health facility and having good knowledge regarding COVID-19 pandemic increase the likelihood of having good practice in utilizing preventive techniques by 2.14 and 1.88 (AOR 2.14 with 95% CI 1.50, 3.06 and 1.88 with 95% CI 1.34, 2.64) respectively. The final factor that affects the utilization of COVID-19 preventive techniques was the underlining myth individuals hold regarding this pandemic. Hence having a wrong myth towards COVID-19 pandemic increases likelihood of having good practice by 1.42(AOR 1.42 with 95% CI 1.02, 1.99) when compared with those without the wrong myth. See "Table 5" below.

## Discussion

Epidemics and pandemics are unexpected periodic phenomena. They can happen at any time. Peoples face several challenges during such conditions. The effects and impacts of pandemics are multiple. It can affect every aspect of life physically, mentally, and emotionally. Hence in this study, we have investigated risk perception, community myth, and practices towards COVID-19 pandemic and factors associated with them. The risk perception was assessed by giving due attention to emotional and knowledge aspects.

This study found as an around half of or 415 (48.6%) of populations have high-risk perceptions. This finding was the same as the finding from one study conducted in Iran [33] but is

**Table 3. Generalized linear model results of factors associated with risk perception towards COVID-19 pandemic in East Bale and Bale Zone Southeast Ethiopia.**

| Variables | Beta | 95 Confidence Interval of beta | | Hypothesis Test | | |
|---|---|---|---|---|---|---|
| | | Lower | Upper | Wald Chi-Square | Df | P-value |
| **Residence** | | | | | | |
| Rural | 0.873 | 0.512 | 1.233 | 22.486 | 1 | 0.000* |
| Urban | 0 | | | | | |
| **Occupation** | | | | | | |
| Gov't employees | 0.102 | -0.511 | 0.716 | 0.107 | 1 | 0.744 |
| NGO employees | 0.533 | -1.198 | 2.265 | 0.364 | 1 | 0.546 |
| Private workers | 0.086 | -0.486 | 0.658 | 0.087 | 1 | 0.768 |
| Farmers | -0.04 | -0.585 | 0.498 | 0.025 | 1 | 0.875 |
| Daily laborers | 0 | | | | | |
| **Distance from HF** | | | | | | |
| Near | 0.393 | | 0.726 | 5.356 | 1 | 0.021* |
| Far | 0 | 0.060 | | | | |
| **History of chronic illness** | | | | | | |
| No | -0.412 | -0.848 | 0.024 | 3.428 | 1 | 0.064 |
| Yes | 0 | | | | | |
| **Wrong myth** | | | | | | |
| No | -0.332 | -0.643 | -0.022 | 4.404 | 1 | 0.036* |
| Yes | 0 | | | | | |
| **Practice** | | | | | | |
| Poor | -0.261 | -0.561 | 0.039 | 2.899 | 1 | 0.089 |
| Good | 0 | | | | | |
| **Knowledge** | | | | | | |
| Poor | -0.216 | -0.536 | 0.104 | 1.750 | 1 | .186 |
| Good | 0 | | | | | |

*Significant factors at P-Value of <0.05, 0 = Reference Category, Df = degree of freedom

higher than the finding from the study conducted in Germany [34] and lower than the finding from the studies conducted in China and Ghana [35, 36]. The disagreement between the current study and studies mentioned could be due to differences in sociodemographic factors like age, residence, educational level and it also might be due to differences in access to information. Also, there is a time difference between when these studies were conducted. The above-mentioned studies were conducted in the early phase of the pandemic. These indicate as level of risk perception in different communities around the world can be different.

This study also found as a rural residence, living near a health facility and having with the wrong myth were significantly and positively associated with risk perception towards the COVID-19 pandemic. This finding conflicts with the studies conducted in Jordan in which urban residence was positively associated with risk perception [37] and it was also not consistent with the other study conducted in China where residence was not significantly associated with risk perception [35, 36]. But this finding was in line with the study conducted in Iran [38].

The reason for the difference could be due to the difference in characteristics of study participants and the speed at which information reach these population based on the status of usage of social media in these different places.

This study also found the prevalence of wrong myths to be 50.1%. This was higher than the finding from a Hospital-based study conducted in Northwest Ethiopia [39]. The reason for

**Table 4. Linear mixed model results of factors associated with community myth towards COVID-19 pandemic in East Bale and Bale Zone Southeast Ethiopia.**

| Variables | Estimate | 95% Confidence Interval | | Df | T-value | P-value |
|---|---|---|---|---|---|---|
| | | Lower | Upper | | | |
| **Gender** | | | | | | |
| Male | -0.053433 | -0.210386 | 0.103520 | 841 | -0.668 | 0.504 |
| Female | 0 | | | | | |
| **Occupation** | | | | | | |
| Gov't worker | -0.248985 | -0.577316 | 0.079346 | 841 | -1.488 | 0.137 |
| NGO worker | 1.466748 | 0.604466 | 2.329031 | 841 | 3.339 | 0.001* |
| Private worker | -0.241984 | -0.548858 | 0.064890 | 841 | -1.548 | 0.122 |
| Farmer | -0.266046 | -0.555788 | 0.023697 | 841 | -1.802 | 0.072 |
| Daily laborer | 0 | | | | | |
| **Residence** | | | | | | |
| Rural | 0.038332 | -0.156203 | 0.232867 | 841 | 0.387 | 0.699 |
| Urban | 0 | | | | | |
| **Distance from HF** | | | | | | |
| Near | 0.006825 | -0.173489 | 0.187140 | 841 | 0.074 | 0.941 |
| Far | 0 | | | | | |
| **Knowledge** | | | | | | |
| Poor | -1.174779 | -1.332594 | -1.016964 | 841 | -14.611 | 0.000* |
| Good | 0 | | | | | |
| **Risk perception** | | | | | | |
| Low | -0.109217 | -0.261076 | 0.042643 | 841 | -1.412 | 0.158 |
| High | 0 | | | | | |
| **Practice** | | | | | | |
| Poor | -.164501 | -.326729 | -.002272 | 841 | -1.990 | |
| Good | 0 | | | | | .047* |

*Significant factors at P-Value of <0.05, 0 = Reference Category, HF = health facility, Df = degree of freedom

disagreement could be the differences in the study setting. Because the current study was community-based while; the study from Northwest Ethiopia was conducted in a health facility which could bring the difference in the finding. The other possible reason could be the time during which these studies were conducted and the difference in the study participants. The study from Northwest Ethiopia was conducted among patients with chronic illness who possibly have regular follow up in the selected Hospital and have a chance to get the right information from health professionals.

This study also identified factors significantly associated with community myths. Accordingly community myth was significantly associated with occupation, knowledge regarding COVID-19, and level of practice regarding utilization of COVID-19 preventive measures. This finding was in line with the finding from the study conducted in South Africa [28] in which knowledge regarding COVID-19 was significantly associated with community myth. But the finding from the current study was not in agreement with the finding from the study conducted in China in which those who have good practice have low myth towards COVID-19 [22]. The reason for the discrepancy could be the difference in sociodemographic characteristics of respondents, the difference in access to information, and the value these communities give for tradition and rumors.

Also, we have found the proportion of people with good practice regarding COVID-19 which is 57.1%. This finding was in line with the other study conducted in Ethiopia [39] where

**Table 5. Generalized linear model results of factors associated utilization of COVID-19 pandemic preventive measures, East Bale and Bale Zone Southeast Ethiopia.**

| Variables | Beta value | Beta 95% Confidence Interval | | Hypothesis Test | | |
| --- | --- | --- | --- | --- | --- | --- |
| | | Lower | Upper | Wald Chi-Square | Degree of freedom | P-value |
| **Gender** | | | | | | |
| Male | -0.324 | -0.667 | 0.018 | 3.454 | 1 | 0.063 |
| Female | 0 | | | | | |
| **Residence** | | | | | | |
| Rural | 0.181 | -0.195 | 0.557 | 0.888 | 1 | 0.346 |
| Urban | 0 | | | | | |
| **Level of education** | | | | | | |
| No formal education | -1.447 | -2.037 | -0.857 | 23.094 | 1 | 0.000* |
| Primary education | -0.773 | -1.289 | -0.257 | 8.634 | 1 | 0.003* |
| Secondary education | -0.164 | -0.731 | 0.402 | 0.322 | 1 | 0.570 |
| College and above | 0 | | | | | |
| **Distance from HF** | | | | | | |
| Near | 0.763 | 0.408 | 1.119 | 17.741 | 1 | 0.000* |
| Far | 0 | | | | | |
| **Substance use** | | | | | | |
| No | 0.321 | -0.020 | 0.662 | 3.412 | 1 | 0.065 |
| Yes | 0 | | | | | |
| **Knowledge** | | | | | | |
| Poor | -0.634 | -0.972 | -0.296 | 13.500 | 1 | 0.000* |
| Good | | | | | | |
| **Risk perception** | | | | | | |
| Low | -0.179 | -0.489 | 0.132 | 1.273 | 1 | 0.259 |
| High | 0 | | | | | |
| **Wrong myth** | | | | | | |
| No | -0.354 | -0.688 | -0.020 | 4.322 | 1 | 0.038* |
| Yes | 0 | | | | | |
| **Age in years** | 0.003 | -0.010 | 0.016 | 0.148 | 1 | 0.701 |

*Significant factors at P-Value of <0.05, 0 = Reference Category, HF = health facility

the proportion of poor practice was 47.3%. This was lower than the finding from studies conducted in Nepal and China [22, 24]. The reason for disagreement could be the difference in sociodemographic characteristics of selected study participants as well as it could be due to the difference in receiving correct information and their access to social media. Higher level of education, living near health facility, with good knowledge about the disease and having wrong myth regarding COVID-19 associated with good practice. This finding was in agreement with other studies conducted in China, Pakistan, and Malaysia [21, 22, 39, 40]. But it was different from what was reported in the study conducted in Sudanese [41]. The reason for the difference between these studies and the current study could be due to the difference in the study setting. Because the current study was conducted in the community and the one from Sudan was an online survey which could bring the difference in study participants. This study identified important findings. But it has limitations. Being a crossectional study this study couldn't identify the direction of association that means whether factors or outcomes come first. It also uses self-report from respondents which could affect the real finding.

## Conclusion

This study is an important step towards a better understanding of risk perception, community myth, and practices regarding COVID-19 pandemics and associated factors. Accordingly, relatively high-risk perception, wrong community myth, and poor practice regarding utilization of COVID-19 preventive techniques were reported. Different factors associated with risk perception, community myth, and practices were identified. These findings could be important input for modeling interventional activities in the community.

## Supporting information

**S1 Questionnaire.**
(DOCX)

**S1 Data.**
(SAV)

## Acknowledgments

Great regards to Madda Walabu University for all support.
Great thanks for study participants.

## Author Contributions

**Conceptualization:** Ahmednur Adem Aliyi, Musa Kumbi Ketaro, Zinash Teferu Engida, Abduljewad Hussen Muhammed, Mesud Mohammed Hassen, Damtow Solomon Shiferaw, Sintayehu Hailu Ayene, Jeylan Kassim Esmael.

**Data curation:** Ahmednur Adem Aliyi, Musa Kumbi Ketaro, Ayele Mamo Argaw, Abduljewad Hussen Muhammed, Abdushekur Mohammed Abduletif, Abate Lette Wodera, Sintayehu Hailu Ayene, Jeylan Kassim Esmael.

**Formal analysis:** Ahmednur Adem Aliyi, Abate Lette Wodera, Jeylan Kassim Esmael.

**Investigation:** Ahmednur Adem Aliyi, Musa Kumbi Ketaro, Zinash Teferu Engida, Ayele Mamo Argaw, Abduljewad Hussen Muhammed, Mesud Mohammed Hassen, Abdushekur Mohammed Abduletif, Damtow Solomon Shiferaw, Abate Lette Wodera, Sintayehu Hailu Ayene, Edao Sinba Etu.

**Methodology:** Mesud Mohammed Hassen, Abdushekur Mohammed Abduletif, Damtow Solomon Shiferaw, Abate Lette Wodera, Sintayehu Hailu Ayene, Edao Sinba Etu.

**Software:** Ahmednur Adem Aliyi, Zinash Teferu Engida, Ayele Mamo Argaw, Edao Sinba Etu.

**Supervision:** Ayele Mamo Argaw.

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
