## [Editor Report · Decision Letter 0]

25 Jan 2022

PONE-D-21-40289Risk perception, community myth, and practices towards COVID-19 pandemic in Southeast Ethiopia: Community based crossectional studyPLOS ONE

Dear Dr. Aliyi,

Thank you for submitting your manuscript to PLOS ONE. After careful consideration, we feel that it has merit but does not fully meet PLOS ONE’s publication criteria as it currently stands. Therefore, we invite you to submit a revised version of the manuscript that addresses the points raised during the review process.

We look forward to receiving your revised manuscript.

Kind regards,

Jong In Kim

Academic Editor

PLOS ONE

Journal Requirements:

" The funders had no role in study design, data collection and analysis, decision to publish, or preparation of the manuscript."

5. Thank you for stating the following in the Funding Section of your manuscript: 

"This study was funded by Madda Walabu University. The funders had no role in study design, data collection, and analysis, or preparation of the manuscript."

" The funders had no role in study design, data collection and analysis, decision to publish, or preparation of the manuscript."

7. Your ethics statement should only appear in the Methods section of your manuscript. If your ethics statement is written in any section besides the Methods, please move it to the Methods section and delete it from any other section. Please ensure that your ethics statement is included in your manuscript, as the ethics statement entered into the online submission form will not be published alongside your manuscript. 

Additional Editor Comments:

This study aims to assess risk perception, community myths, and preventive practice towards COVID-19 in Ethiopia, 2020.

A community-based cross-sectional study was conducted among 854 participants selected using a multistage sampling technique.

From 854 respondents included in this study, 428 (50.1%) have a wrong tale about COVID-19, and 366 (42.9%) have poor practice, respectively. Thus, the study lays out high-risk perceptions, false community myths, and relatively low utilization of available methods towards COVID-19 and their associated factors.

1. On time, I evaluate it is good to investigate the awareness level of the country, in informing the international community that has a wrong tale about COVID-19.

2. Please supplement the discussion further to reinforce the direction of improvement for health promotion against problems of COVID-19 in Ethiopia.

See also the following paper:

In Kim, J., Kim, G. & Choi, Y. Effects of air pollution on children from a socioecological perspective. BMC Pediatr 19, 442 (2019). https://doi.org/10.1186/s12887-019-1815-x

Kim, J.I., Kim, G. Effects on inequality in life expectancy from a social ecology perspective. BMC Public Health 18, 243 (2018). https://doi.org/10.1186/s12889-018-5134-1
---

## [Author Response · Author response to Decision Letter 0]

14 Mar 2022

RESPONSE TO REVIEWERS

Thank you for reading and commenting our manuscript. We have made changes as per given comments. The responses were provided in the following table. 

Thank you again

COMMENTS RESPONSES 

Response: Thank you for your comment. We have revised our manuscript as per the requirements. Thank you again

2. Please review your refer ence list to ensure that it is complete and correct.

Response: Thank you for your comment. We have revised our references but we couldn’t find problem. Thank you again

Response: Thank you for your comments. We have thoroughly edit the manuscript to make it clear and informative. We have highlighted bold where changes were made. Thank you again.

" The funders had no role in study design, data collection and analysis, decision to publish, or preparation of the manuscript."

 Response: Thank you for your comments. We have made changes accordingly. Thank you againn 

5. Thank you for stating the following in the Funding Section of your manuscript: 

"This study was funded by Madda Walabu University. The funders had no role in study design, data collection, and analysis, or preparation of the manuscript."

" The funders had no role in study design, data collection and analysis, decision to publish, or preparation of the manuscript."

 Response: Thank you for your comments. We have incorporated this comments and make modification accordingly. We have have removed the funding information from acknowledgment section

6. In your Data Availability statement, you have not specified where the minimal data set underlying the results described in your manuscript can be found 

Response: Thank you for your comments. We have submitted minimal data as supporting information along with this submission

 Thank you again

7. Your ethics statement should only appear in the Methods section of your manuscript 

Response: Thank you for your comment. We made it accordingly and write ethics statement on only methods section. Thank you again.

---

## [Editor Report · Decision Letter 1]

23 Mar 2022

PONE-D-21-40289R1Risk perception, community myth, and practices towards COVID-19 pandemic in Southeast Ethiopia: Community based crossectional studyPLOS ONE

Dear Dr. Aliyi,

Thank you for submitting your manuscript to PLOS ONE. After careful consideration, we feel that it has merit but does not fully meet PLOS ONE’s publication criteria as it currently stands. Therefore, we invite you to submit a revised version of the manuscript that addresses the points raised during the review process.

We look forward to receiving your revised manuscript.

Kind regards,

Jong In Kim

Academic Editor

PLOS ONE

Journal Requirements:

Additional Editor Comments (if provided):

Please supplement the discussion further to reinforce the direction of improvement for health promotion against problems of COVID-19 in Ethiopia.
---

## [Author Response · Author response to Decision Letter 1]

30 Mar 2022

RESPONSE TO REVIEWERS

Thank you for reading and commenting our manuscript. We have made changes as per given comments. The responses were provided in the following table. 

Thank you again

COMMENTS RESPONSES 

Thank you for your comment. We have revised our manuscript as per the requirements. Thank you again

2. Please review your refer ence list to ensure that it is complete and correct.

Thank you for your comment. We have revised our references but we couldn’t find problem. Thank you again

3. We suggest you thoroughly copyedit your manuscript for language usage, spelling, and grammar. If you do not know anyone who can help you do this, you may wish to consider employing a professional scientific editing service. Thank you for your comments. We have thoroughly edits the manuscript to make it clear and informative. We have highlighted bold where changes were made. Thank you again.

" The funders had no role in study design, data collection and analysis, decision to publish, or preparation of the manuscript."

 Thank you for your comments. We have made changes accordingly. Thank you again 

5. Thank you for stating the following in the Funding Section of your manuscript: 

"This study was funded by Madda Walabu University. The funders had no role in study design, data collection, and analysis, or preparation of the manuscript."

" The funders had no role in study design, data collection and analysis, decision to publish, or preparation of the manuscript."

 Thank you for your comments. We have incorporated these comments and make modification accordingly. We have removed the funding information from acknowledgment section

6. In your Data Availability statement, you have not specified where the minimal data set underlying the results described in your manuscript can be found Thank you for your comments. We have submitted minimal data as supporting information along with this submission

 Thank you again

7. Your ethics statement should only appear in the Methods section of your manuscript Thank you for your comment. We made it accordingly and write ethics statement on only methods section. Thank you again.

Please review your reference list to ensure that it is complete and correct. If you have cited papers that have been retracted, please include the rationale for doing so in the manuscript text, or remove these references and replace them with relevant current references. Any changes to the reference list should be mentioned in the rebuttal letter that accompanies your revised manuscript. If you need to cite a retracted article, indicate the article’s retracted status in the References list and also include a citation and full reference for the retraction notice. Thank you for your comment. We have updated all references for their completeness. And we have remove references used mistakenly before their publication (preprint).

Thank you again

---

## [Decision Letter · Decision Letter 2]

15 Aug 2022

PONE-D-21-40289R2Risk perception, community myth, and practices towards COVID-19 pandemic in Southeast Ethiopia: Community based crossectional studyPLOS ONE

Dear Dr. Aliyi

Thank you for submitting your manuscript to PLOS ONE. After careful consideration, we feel that it has merit but does not fully meet PLOS ONE’s publication criteria as it currently stands. Therefore, we invite you to submit a revised version of the manuscript that addresses the points raised during the review process.

We look forward to receiving your revised manuscript.

Kind regards,

Soham Bandyopadhyay

Academic Editor

PLOS ONE

Journal Requirements:

Reviewers' comments:

Reviewer's Responses to Questions

**Comments to the Author**

1. If the authors have adequately addressed your comments raised in a previous round of review and you feel that this manuscript is now acceptable for publication, you may indicate that here to bypass the “Comments to the Author” section, enter your conflict of interest statement in the “Confidential to Editor” section, and submit your "Accept" recommendation.

Reviewer #1: (No Response)

Reviewer #2: (No Response)

2. Is the manuscript technically sound, and do the data support the conclusions?

Reviewer #1: Yes

Reviewer #2: Yes

3. Has the statistical analysis been performed appropriately and rigorously? 

Reviewer #1: Yes

Reviewer #2: I Don't Know

4. Have the authors made all data underlying the findings in their manuscript fully available?

Reviewer #1: Yes

Reviewer #2: Yes

5. Is the manuscript presented in an intelligible fashion and written in standard English?

Reviewer #1: Yes

Reviewer #2: Yes

6. Review Comments to the Author

Reviewer #1: (No Response)

Reviewer #2: The manuscript adds useful information about the risk perception, community myth, and practices towards COVID-19 pandemic in the Africa setting. I have the following specific comments for the authors' consideration:

a. The introduction section of the manuscript needs revision with emphasis on style of English. Some of the sentences are not written in standard English.

b. Line 51 - replace “.” and “And” with ","

c. Line 67 - replace "can transmit" with “can also be transmitted”

d. Line 200 - remove “shows”, include "N" and remove "female" from table 1

7. PLOS authors have the option to publish the peer review history of their article (what does this mean?). If published, this will include your full peer review and any attached files.

Reviewer #1: **Yes: **Mohammedaman Mama Hussen

Reviewer #2: **Yes: **Ms. Catherine Okoi

---

## [Author Response · Author response to Decision Letter 2]

18 Aug 2022

Comments to the Author

1. If the authors have adequately addressed your comments raised in a previous round of review and you feel that this manuscript is now acceptable for publication, you may indicate that here to bypass the “Comments to the Author” section, enter your conflict of interest statement in the “Confidential to Editor” section, and submit your "Accept" recommendation.

Reviewer #1: (No Response)

Reviewer #2: (No Response)

Thank for reviewing our paper by giving your precious time. Thank you again

2. Is the manuscript technically sound, and do the data support the conclusions?

Reviewer #1: Yes

Reviewer #2: Yes

Thank for reviewing our paper by giving your precious time. Thank you again

3. Has the statistical analysis been performed appropriately and rigorously? 

Reviewer #1: Yes

Reviewer #2: I Don't Know

Thank for reviewing our paper by giving your precious time. Thank you again

4. Have the authors made all data underlying the findings in their manuscript fully available?

Reviewer #1: Yes

Reviewer #2: Yes

Thank for reviewing our paper by giving your precious time. Thank you again

5. Is the manuscript presented in an intelligible fashion and written in standard English?

Reviewer #1: Yes

Reviewer #2: Yes

6. Review Comments to the Author

Reviewer #1: (No Response)

Reviewer #2: The manuscript adds useful information about the risk perception, community myth, and practices towards COVID-19 pandemic in the Africa setting. I have the following specific comments for the authors' consideration:

a. The introduction section of the manuscript needs revision with emphasis on style of English. Some of the sentences are not written in standard English.

b. Line 51 - replace “.” and “And” with ","

c. Line 67 - replace "can transmit" with “can also be transmitted”

d. Line 200 - remove “shows”, include "N" and remove "female" from table 1

7. PLOS authors have the option to publish the peer review history of their article (what does this mean?). If published, this will include your full peer review and any attached files.

Thank for reviewing our paper by giving your precious time. We have incorporated the comment raised in the tack changes manuscript. Thank you again

Do you want your identity to be public for this peer review? For information about this choice, including consent withdrawal, please see our Privacy Policy.

Reviewer #1: Yes: Mohammedaman Mama Hussen

Reviewer #2: Yes: Ms. Catherine Okoi

Thank for reviewing our paper by giving your precious time. Thank you again

---

## [Editor Report · Decision Letter 3]

14 Sep 2022

Risk perception, community myth, and practices towards COVID-19 pandemic in Southeast Ethiopia: Community based crossectional study

PONE-D-21-40289R3

Dear Dr.  Aliyi

We’re pleased to inform you that your manuscript has been judged scientifically suitable for publication and will be formally accepted for publication once it meets all outstanding technical requirements.

Kind regards,

Soham Bandyopadhyay

Academic Editor

PLOS ONE

---

## [Editor Report · Acceptance letter]

21 Sep 2022

PONE-D-21-40289R3 

Risk perception, community myth, and practices towards COVID-19 pandemic in Southeast Ethiopia: Community based crossectional study 

Dear Dr. Aliyi:

I'm pleased to inform you that your manuscript has been deemed suitable for publication in PLOS ONE. Congratulations! Your manuscript is now with our production department. 

Kind regards, 

on behalf of

Dr. Soham Bandyopadhyay 

Academic Editor

PLOS ONE